# Heterologous Expressed NbSWP12 from Microsporidia *Nosema bombycis* Can Bind with Phosphatidylinositol 3-Phosphate and Affect Vesicle Genesis

**DOI:** 10.3390/jof8080764

**Published:** 2022-07-23

**Authors:** Jie Chen, Zhi Li, Xiaotian Sheng, Jun Huang, Quan Sun, Yukang Huang, Rong Wang, Yujiao Wu, Mengxian Long, Jialing Bao, Zeyang Zhou, Guoqing Pan

**Affiliations:** 1State Key Laboratory of Silkworm Genome Biology, Southwest University, Chongqing 400716, China; jchen@swu.edu.cn (J.C.); sxt9929@outlook.com (X.S.); hjunjun114@163.com (J.H.); quan_suen@163.com (Q.S.); hyk1990@126.com (Y.H.); wangrong199501@126.com (R.W.); wuyujiao523@163.com (Y.W.); longmx@swu.edu.cn (M.L.); baojl@swu.edu.cn (J.B.); 2Chongqing Key Laboratory of Microsporidia Infection and Control, Southwest University, Chongqing 400716, China; 3College of Life Sciences, Chongqing Normal University, Chongqing 401331, China; lizhi.cqnu@gmail.com

**Keywords:** microsporidia, *Nosema bombycis*, PIP3-binding protein, vesicle genesis, spore wall protein

## Abstract

Microsporidia are a big group of single-celled obligate intracellular organisms infecting most animals and some protozoans. These minimalist eukaryotes lack numerous genes in metabolism and vesicle trafficking. Here, we demonstrated that the spore wall protein NbSWP12 of microsporidium *Nosema bombycis* belongs to Bin/Amphiphysin/Rvs (BAR) protein family and can specifically bind with phosphatidylinositol 3-phosphate [Ptdlns(3)P]. Since Ptdlns(3)P is involved in endosomal vesicle biogenesis and trafficking, we heterologous expressed NbSWP12 in yeast *Saccharomyces cerevisiae* and proved that NbSWP12 can target the cell membrane and endocytic vesicles. *Nbswp12* transformed into *Gvp36* (a BAR protein of *S. cerevisiae*) deletion mutant rescued the defect phenotype of vesicular traffic. This study identified a BAR protein function in vesicle genesis and sorting and provided clues for further understanding of how microsporidia internalize nutrients and metabolites during proliferation.

## 1. Introduction

Microsporidia constitute a phylum (Microspora) of obligate intracellular parasites related to fungi. In recent years, the harmness of Microsporidiosis in humans, livestock, and aquaculture, as well as laboratory animal infection, has become increasingly prominent [1,2]. Blocking the proliferation of microsporidia within host cells may be an important strategy for prevention and control. As single-celled microorganisms parasitic inside the host cells, genomes of microsporidia have been extremely reduced. Metabolic pathways such as tricarboxylic acid cycle, fatty acid *β*-oxidation, and de novo synthesis of amino acids that most of the eukaryotes relied on have been lost in microsporidia [3]. To achieve their proliferation and development, transporters or endocytosis are used by microsporidia to steal energy and metabolites from host cells [1,4]. *Nosema bombycis*, the earliest named microsporidium, is the pathogen of the silkworm *Bombyx mori* and has been isolated from several lepidoptera [5,6]. Several key genes of the vesicle transport pathway have been identified in the genome of *N. bombycis* [7]. However, how endocytosis occurs in the microsporidie and how the plasma membrane recycles during endocytosis is still unknown.

The interactions between lipids and proteins e for cells regulate membrane curvature, and it is also the premise to ensure the process of cell molding, vesicle formation, and autophagy [8]. It is known that the Bin/Amphiphysin/Rvs (BAR) domain protein acts as a sensor and inducer of cell membrane curvature, and its binding to lipids plays an important role in cell membrane shaping and vesicle genesis [9,10,11,12]. The formation of endocytic vesicles is dependent on the production of phosphoinositide (PI) and its interaction with various endocytosis-related proteins [13]. Phosphatidylinositols such as phosphatidylinositol 3-phosphate [Ptdlns(3)P], phosphatidylinositol 4,5-bisphosphate [PtdIns(4,5)P_2_], phosphatidylinositol-3,4-bisphosphate [PtdIns(3,4)P_2_], and phosphatidylinositol (3,4,5)-trisphosphate [PtdIns(3,4,5)P_3_] are all involved in the formation or transport of vesicles. Ptdlns(3)P or PI3P can accumulate in the membrane where endosomes are generated [13]. Ptdlns(3)P and PtdIns(4,5)P_2_ modulate membrane curvature by selectively recruiting BAR domain protein, such as Snx9, to alter actin polymerization [14].

Currently, the only proteins predicted to have a BAR domain reported in microsporidia are NbSWP12 from *N. bombycis* and its homologous proteins [15,16]. NbSWP12 was originally reported in the proteomic study of the spore wall of *N. bombycis* [17]. Subsequent studies found that NbSWP12 was not only localized to the spore wall but also abundantly expressed in the sporoplasm and meront stages during *N. bombycis* infection and accumulated at the poles of the elongated meronts during division [18,19]. RNA interference and transgenic cells stably expressing single-chain antibodies against NbSWP12 could effectively inhibit the proliferation of *N. bombycis* [18]. Therefore, we are very interested to know how NbSWP12 functions as a BAR protein and if it has the potential to participate in the endocytosis process.

To overcome the known technical difficulties of performing genetic modifications on microsporidia, we utilized the yeast *Saccharomyces cerevisiae* in this study for genetic modification and protein function analysis. We demonstrated that NbSWP12 had two features of the BAR family, dimerization and lipid-binding ability. Lipid overlay assays demonstrated that NbSWP12 specifically binds to Ptdlns(3)P. Particularly, NbSWP12 expressed in the yeast can affect vesicle genesis and rescue the vacuole biogenesis defect of *Saccharomyces cerevisiae* BAR protein gvp36 deletion mutant. Our findings open a door for further study of how microsporidia internalize nutrients and metabolites during proliferation.

## 2. Materials and Methods

### 2.1. Parasite, Cells, and Cell Culture

Microsporidia *Nosema bombycis* CQI was isolated from infected silkworm *Bombyx mori* in Chongqing, China, and conserved in the China Veterinary Culture Collection Center (CVCC no. 102059). The mature spores of *N. bombycis* were purified by discontinuous Percoll gradient centrifugation (30%, 45%, 60%, 75%, and 90% (*v*/*v*)) and centrifuged at 40,000× *g* for 30 min. The washed spores were stored in ddH_2_O supplemented with antibiotics (Penicillin-Streptomycin solution, 100×, Beyotime, Haimen, China) at 4 °C. The purified parasites were grown in Sf9 cells that were cultured in Sf-900 III SFM medium (Gibco, New York, NY, USA) at 28 °C.

The *Saccharomyces cerevisiae* strain CEN.PK2 (*MAT a/α*, *ura3-52*, *leu2-3,112*, *trp1-289*, and *his3-1*) and plasmid pUG35 were kindly provided by Dr. Johannes H. Hegemann (Heinrich-Heine-Universität, Düsseldorf, Germany), which were used for targeting protein in yeast [20]. The BY4742 (*MATα, his3*Δ*1, leu2*Δ*0, lys2*Δ*0*, and *ura3*Δ*0*) wild-type and *gvp36*Δ knockout strain of *S. cerevisiae* were purchased from Open Biosystems (Thermo Fisher Scientific, Huntsville, AL, USA).

### 2.2. Multiple Sequence Alignment

Sequences of *swp12* homologous genes from 16 microsporidia and *gvp36* from *Saccharomyces cerevisiae* were downloaded from the public database NCBI (https://www.ncbi.nlm.nih.gov/ (accessed on 16 January 2022)). The sequences were aligned with Clustal Omega (https://www.ebi.ac.uk/Tools/msa/ (accessed on 16 January 2022)) and rendered by ESPript 3.0 [2].

### 2.3. Protein Expression and Purification

The previously constructed *Escherichia coli* Rosetta cells, which carry p-Cold I-*Nbswp12* plasmids [3], were cultured in a Luria-Broth (LB) medium containing 100 μg/mL ampicillin at 37 °C with shaking to an optical density (OD 600 nm) of 0.6. Then recombinant protein was induced with 0.2 mM Isopropyl *β*-D-1-thiogalactopyranoside (IPTG) for 20 h at 16 °C. Bacterial cells were harvested by centrifugation at 8000× *g* for 5 min and suspended in Binding buffer (20 mM sodium phosphate, 500 mM NaCl, 20 mM imidazole, pH7.4), followed by sonication. The recombinant protein was purified with HisTrap HP and Hi Load 16/60 Superdex 75 (GE Health), following the manufacturer’s instructions. Purified protein was eluted with 10 mM Tris–HCl (pH 8.0), quantified with Bradford Protein Assay Kit (Beyotime), and stored at −20 °C.

### 2.4. Non-Reducing SDS-PAGE

Briefly, 12% SDS-PAGE gel was prepared for this study. Purified recombinant NbSWP12 (rNbSWP12) protein was equally divided into two samples. One sample mixed with non-reducing loading buffer (30 mL 0.3 mol/L Tris-HC1(pH 6.8), 0.075 g Bromophenol blue, 12 mL Glycerol, dissolved in ddH_2_O up to 50 mL). The other mixed with SDS-PAGE sample loading buffer (30 mL 0.3 mol/L Tris-HC1(pH 6.8), 0.075 g Bromophenol blue, 12 mL Glycerol, 3 g SDS and 6 mL β-mercaptoethanol, dissolved in ddH_2_O up to 50 mL) and incubated at 98 °C for denaturation. Separated proteins were transferred to the PVDF membrane, and immunoblotting against NbSWP12 with antiserum (dilution in 1:5000) was performed to analyze the monomer or dimer of NbSWP12.

### 2.5. Yeast Two-Hybrid Analysis

*Nbswp12* gene sequence is 684 bp in length and encodes 228 amino acids. For this study, DNA fragment encoding NbSWP12 was cloned from p-Cold I-*Nbswp12* plasmid generated previously [15] and subcloned into pGADT7 and pGBKT7 plasmids (Clontech) using restriction endonucleases (TAKARA) to create the GAL4 DNA-activation and DNA-binding domain fusions, respectively. The list of primers is available in Table 1. The recombinant plasmids were transformed into the *Saccharomyces cerevisiae* AH109 or Y187 strain (Clontech), respectively, by Yeast Transformation Kit (Clontech). Yeast transformants were selected on synthetic dropout medium (SD) plates lacking leucine or tryptophan (Clontech). To validate protein-protein interactions, a selective medium SD/-Leu-Trp-His-Ade supplemented with X-α-Gal was used. The fusion strain of pGBKT7-53 with pGADT7-T was used as the positive control, while the fusion strain of pGBKT7-lam with pGADT7-T, the fusion strain of pGBKT7-53 with pGADT7-*Nbswp12* and the fusion strain of pGBKT7-*Nbswp12* with pGADT7-T were used as negative controls.

### 2.6. Liposome Co-Sedimentation

Folch fractions from bovine brain purchased from Sigma were dissolved in methanol–chloroform mixture (*v*/*v* = 3:1) to reach the concentration of 10 mg/mL and dried by a rotavapor. Then, 20 mmol/L Hepes buffer (20 mmol/L Hepes, pH 7.4, 150 mmol/L NaCl, 1 mmol/L DTT) was applied to resuspend dried lipids, incubated for 15 min at room temperature, followed by 10 passages through Millipore 0.22 or 0.45 mm diameter polycarbonate filter. Then, 0.05 µg/µL rNbSWP12 was incubated in the presence or absence of 1 µg/µL 0.22 or 0.45 µm Syringe-driven Filter extruded liposomes for 15 min at room temperature. Samples were then centrifuged at conditions of 4 °C, 30,000× *g* for 10 min and analyzed by SDS–PAGE as supernatant and three washed pellet fractions.

### 2.7. Lipid Strip Assay

PIP Strips™ membranes (Molecular Probes) spotted with 100 pmol samples of 15 different phospholipids, and a blank sample were purchased, and a protein-lipid overlay assay was performed following the instruction of the manufacturer. Briefly, the membrane was blocked with TBST (10 mM Tris–HCl, pH 8.0, 150 mM NaCl, containing 0.1% (*v*/*v*) Tween 20) containing 3% fatty acid-free bovine serum albumin (BSA) for an hour at room temperature and incubated with 0.5 μg/mL recombinant NbSWP12 protein in TBST-3% BSA for 1 h at room temperature. The membrane was washed three times and then analyzed by immunoblotting with NbSWP12 antiserum.

### 2.8. Targeting of NbSWP12 in Saccharomyces cerevisiae

For GFP tagging of NbSWP12, DNA fragment of *Nbswp12* was amplified by PCR from p-Cold I-*Nbswp12* plasmid with primers containing *Bam*H I or *Sal* I restriction site (Table 1) and inserted into pUG35 plasmid. The pUG35 and pUG35-*Nbswp12* plasmids were transformed into yeast *S. cerevisiae* CEN.PK2 using Yeastmaker™ Yeast Transformation System 2 (Clontech, Mountain View, CA, USA). Transformants were selected in Minimal synthetic defined (SD) bases with Uracil Dropout Supplement (SD/-Ura). Isolated clone was grown in SD/-Ura liquid overnight and inoculated into SD/-Ura-Met inducing medium for overexpression under control of MET25 inducible promoter. Yeasts were collected at 24 or 48 h post inoculating, and location signals were observed under Olympus FV1200 Laser Scanning Microscope.

### 2.9. Vacuole Staining

The yeast cells were cultured to the mid-logarithmic growth phase in SD/-Met liquid medium. The cultures were centrifuged at 5000 rpm for 2 min and resuspended in a fresh medium containing 80 μM FM4-64 (Invitrogen) for vacuoles staining. Yeasts were centrifuged after 1 h of incubation, resuspended in the fresh medium again, and cultured for extra 3 h. Yeast cells were then sampled on glass slides and observed under Olympus FV1200 Laser Scanning Microscope.

### 2.10. Complementation of Yeast gvp36Δ by NbSWP12

Plasmid pUG35 or pUG35-*Nbswp12* was transformed into wild-type BY4742 or *gvp36*Δ, and the transformed yeast cells were selected on SD/-Ura plates. A single colony of recombined yeast and wild-type BY4742 or *gvp36*Δ was inoculated in YPDA or SD/-Met (Clontech) liquid medium. The overnight cultures were centrifuged, and yeasts were stained with 80 μM FM4-64 as descript above for vacuoles staining. Complementation of defection in vacuole biogenesis was observed under Olympus FV1200 Laser Scanning Microscope.

## 3. Results

### 3.1. SWP12 and S. cerevisiae BAR Protein Gvp36 Possess Two Representative Conserved Motifs

SWP12 homologues contained predicted Bin-amphiphysin-Rvs-2 (BAR-2) domain [16]. The previous analysis showed two motifs YEH/NGG (N refers to neutral amino acids) and RYDLE were conserved in SWP12 of microsporidia [15]. Gvp36 is a BAR protein involved in vesicular traffic and nutritional adaptation in *S. cerevisiae* [21]. From the sequence alignment among SWP12 homologs in microsporidia and the BAR-2 domain of Gvp36 displayed, the two motifs were also relatively conserved in the sequences (Figure 1), indicating similar functions.

### 3.2. NbSWP12 Forms Homodimer

BAR family proteins frequently have a dimerization and lipid-binding domain [10]. To determine whether NbSWP12 (BAR domain e-value: 1.35 e-03) can dimerize, prokaryotic and eukaryotic expressed NbSWP12 were analyzed. Purified His_6_-NbSWP12, which was expressed in *E. coli* Rosetta, were separated by Non-reduced SDS-PAGE. The sample was treated with *β*-mercaptoethanol (*β*-ME), and a high temperature was used to reduce the protein to a monomer. As shown in Figure 2A, the NbSWP12 antibody can detect two bands near ~30 kDa and ~60 kDa, which were consistent with the size of the predicted monomer and dimer of His6-NbSWP12. After *β*-ME and 98 °C treatment, only monomers can be detected. In addition, a yeast two-hybrid assay was performed using pGADT7-*Nbswp12* and pGBKT7-*Nbswp12* to further determine the in vivo dimerization of NbSWP12. The results showed NbSWP12 can interact with itself, then activate the expression of downstream reporter genes. In general, NbSWP12 can form dimers, which is one of the key features of BAR proteins.

### 3.3. NbSWP12 Binds to Ptdlns(3)P

Since NbSWP12 contains a predicted BAR domain, which is a well-known dimerization, membrane-binding, and curvature-sensing module [10], an in vitro liposome binding assay was performed to test the lipid-binding property. As shown in Figure 3A, NbSWP12 recombinant proteins (rNbSWP12) can bind with 0.22 µm or 0.45 µm Folch fraction-derived liposomes in the co-sedimentation assay. To further identify the properties of the interacting lipids, we performed a protein-lipid overlay assay. rNbSWP12 (0.5 µg/mL) were incubated with PIP Strips™ membrane spotted with 15 different phospholipids. Binding between rNbSWP12 and phospholipids was confirmed by immunoblotting with antibodies against NbSWP12. The rNbSWP12 can notably bind to Ptdlns(3)P and weakly interact with phosphatidic acid and other phosphatidylinositol phosphates Ptdlns(4)P, Ptdlns(5)P, Ptdlns(3,5)P_2_, Ptdlns(4,5)P_2_, and Ptdlns(3,4,5)P_3_.

### 3.4. NbSWP12 Targets to Yeast Cell Membrane and Endocytic Vesicle

As genetic modification techniques have not been successful in microsporidia, we used yeast as a substitute species for further study of the protein. The pUG35-*Nbswp12* plasmids were transformed into yeast *S. cerevisiae* CEN.PK2, then NbSWP12-GFP, was expressed. Methionine-deficient medium promoted overexpression of NbSWP12-GFP. As shown in Figure 4A, we observed NbSWP12-GFP fusion protein located in yeast cell membrane. It was consistent with the localization in *N. bombycis* [18]. After 24 h overexpression, NbSWP12-GFP starts to show spots signal looks such as endocytic vesicles in the cells. As with FM4-64 co-staining, cells with low expression of NbSWP12-GFP showed a large single vacuole in the mother and daughter cells. While cells contained much more over-expressed NbSWP12-GFP exhibited numerous vesicles, indicating that vesicles were forming with the aid of NbSWP12, and leading the vacuoles were not well formed. e. GFP was over-expressed throughout the cytoplasm of the yeast, and large vacuoles were formed in the cells.

### 3.5. NbSWP12 Rescues the Vesicle Genesis Defect of Yeast gvp36Δ

To evaluate whether NbSWP12 can affect vesicle genesis, NbSWP12 was expressed in the *S. cerevisiae gvp36*Δ mutant strain to assess whether NbSWP12 can rescue the defective phenotype of vacuole biogenesis. Considering overexpression of NbSWP12 in yeast can cause the formation of numerous vesicles, strains were grown in a YPD medium to ensure regular expressions. Lipophylic dye FM4-64 was used to stain vacuoles by 1 h of staining and 3 h of cultivation with cells. As shown in Figure 5, the vesicles in wild-type yeast cells can transport from the cell membrane and finally form a large vacuole. *gvp36*Δ exhibited several small vacuoles as the mutant defected in membrane trafficking and in the initial steps of endocytosis [21]. Expression of NbSWP12 in *gvp36*Δ showed a single large vacuole in the cells, indicating that NbSWP12 rescued the vesicle genesis defect in yeast *gvp36*Δ.

## 4. Discussion

Microsporidia rely on the host cells’ metabolism and organelles to facilitate obligate intracellular parasitic life cycles [23,24,25,26]. As a minimalist eukaryote, microsporidia lost numerous genes and several organelles during the adaption of parasitism [27,28,29]. The smart organism used a variety of strategies such as directly interacting with the cytoplasm of the host cell, transporting energy and metabolites from the host, and secreting molecules to regulate the host cell [1,4,26,30,31,32]. Endocytosis and vesicle transport may be significant for microsporidia to acquire and utilize macromolecular materials from the host. Sections of microsporidia during infection and intracellular stages showed a large number of membrane structures [1]. Comparative genomic analyses identify clathrin vesicle transport machinery is conserved in primitive microsporidia of the Metchnikovellidae family [33]. Genome research indicated that highly derived microsporidia contained an extremely reduced endosomal system [34,35,36]. Nevertheless, several genes such as adaptin *AP1*, clathrin do exist in genome of some microsporidia such as *Amphiamblys* sp. (accession: XP_013237374.1), *Encephalitozoon cuniculi* (accession: CAD25437.1), and *Nosema bombycis* (accession: EOB15421.1), suggesting a prominent reduction endocytic pathway in microsporidia [7,33,34,37]. The biological processes of microsporidium regulating the formation, anchorage, and fusion of vesicles may be different from those of other eukaryotes.

In this study, we demonstrated a new BAR protein, NbSWP12, which can bind with Ptdlns(3)P and affect vesicle genesis in yeast. NbSWP12 was firstly identified as a spore wall protein in microsporidia *Nosema bombycis* [15,17]. Subsequent studies indicated it is expressed throughout the life cycle of *N. bombycis* and located at the membrane of meronts, concentrated at both ends of the elongated meronts during division [18,19]. BAR proteins can be a sensor and regulator of membrane curvature. During the division of microsporidian meronts, BAR proteins may interact with lipids on the membrane and recruit other proteins, which jointly act on the polarization and division of the meronts. The aggregation location of NbSWP12 at the two poles of the meronts suggested that NbSWP12 may participate in the processes as a BAR protein. Since PtdIns(3)P mostly participated in the endosome formation and distribution, endosomal fusion, membrane trafficking and sorting, autophagy, as well as signaling, the interaction between NbSWP12 and PtdIns(3)P further confirmed the function of NbSWP12 as BAR protein [38,39,40,41,42]. The membrane and vesicle localization of NbSWP12 in yeast cells and the vacuole biogenesis defect rescue experiment indicated that NbSWP12 participates in the endosomal pathway.

PtdIns(3)P and its cooperated proteins play a critical regulatory role in cargo sorting. As we can see from the yeast location signal, NbSWP12 is located in particular vesicles, suggesting these vesicles may be involved in cargo sorting. The intracellular proliferation of microsporidia requires a large number of metabolites derived from sugars, lipids, amino acids, and other growth-related factors from the host cytoplasm [25,43,44]. What substances are internalized by the endosomes that NbSWP12 participates in forming, and where is the destination? What is the molecular mechanism behind the formation of this specific endocytosis? These questions are of great importance and worthy of further studies in the future.

## Figures and Tables

**Figure 1 jof-08-00764-f001:**
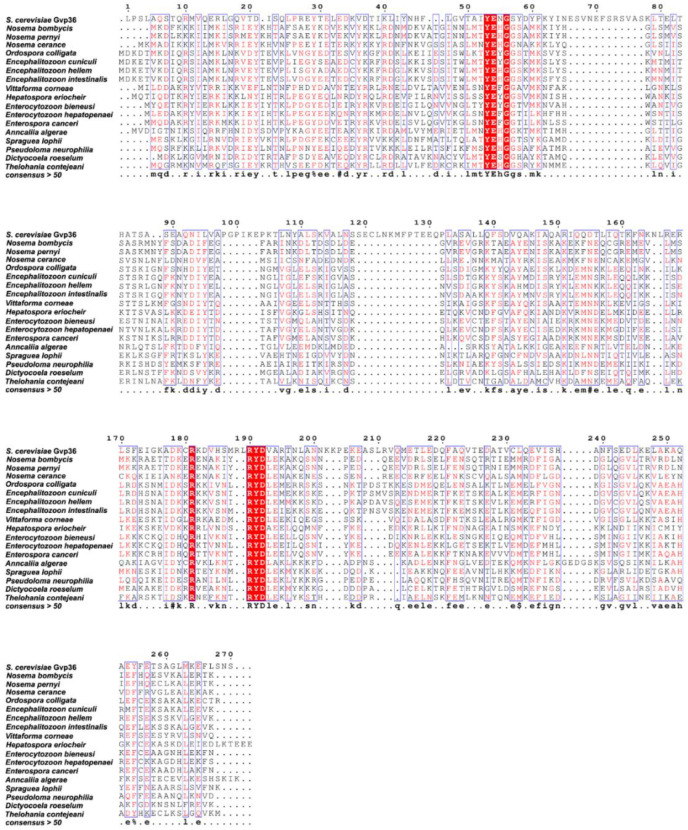
Multiple sequence alignment of microsporidian SWP12 homologs and the BAR-2 domain of *Saccharomyces cerevisiae* Gvp36. The sequences were aligned with Clustal Omega and rendered by ESPript 3.0. The colored amino acids with blue frames represented the consensus level is greater than 0.7. White characters on a red background highlight the strictly conserved residues. A consensus sequence is generated using criteria from MultAlin. Uppercase is identity, lowercase is consensus level > 0.7, $ is anyone of LM, % is anyone of FY, and # is anyone of NDQEBZ [22].

**Figure 2 jof-08-00764-f002:**
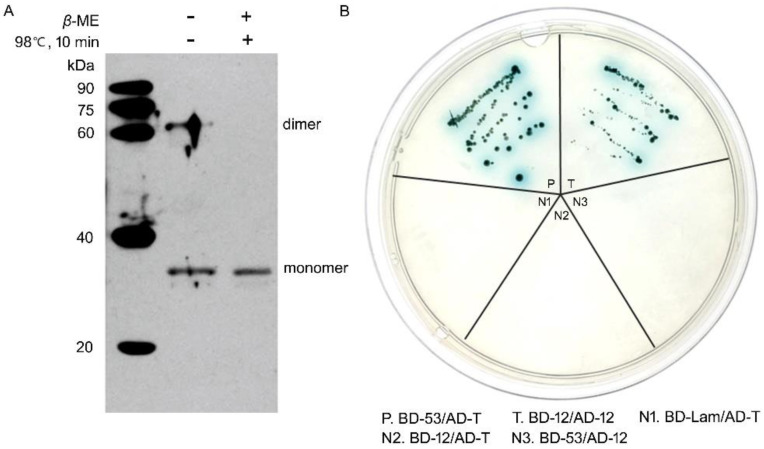
Heterologous expressed NbSWP12 can form homodimer. (**A**) Non-reduced SDS-PAGE combined with Western blotting analysis of His_6_-NbSWP12 recombinant protein expressed in *Escherichia coli* Rosetta. The NbSWP12 specific antibody can recognize monomer and dimer of NbSWP12. *β*-ME: *β*-mercaptoethanol. (**B**) A yeast two-hybrid assay was used to further determine the in vivo dimerization of NbSWP12. P, the fusion strain of pGBKT7-53 with pGADT7-T was used as the positive control. T, the fusion strain of pGBKT7-*Nbswp12* with pGADT7- *Nbswp12* was used as test group. The fusion strain of pGBKT7-lam with pGADT7-T (N1), the fusion strain of pGBKT7-*Nbswp12* with pGADT7-T (N2), and the fusion strain of pGBKT7-53 with pGADT7-*Nbswp12* (N3) were used as negative controls. The fusion strain in test group can grow on SD-Leu-Trp-His-Ade/ X-α-Gal medium and generate blue metabolites, indicating NbSWP12 can interact with each other and form homodimer.

**Figure 3 jof-08-00764-f003:**
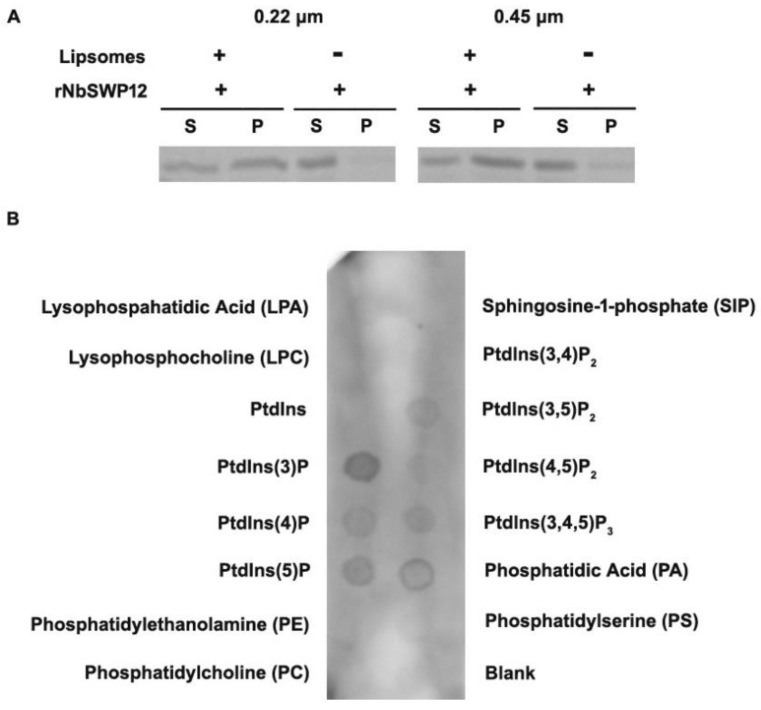
NbSWP12 interacts with phosphoinositide. (**A**) SDS-PAGE analysis of NbSWP12 and liposomes interaction; 0.05 µg/µL bacterial purified rNbSWP12 incubated with 1 μg/μL 0.22 µm or 0.45 µm Folch fraction derived liposomes in a cosedimentation assay. Supernatant (S) and precipitate (P) were loaded and separated on 12% SDS-PAGE. The results indicated that rNbSWP12 binds with lipids. (**B**) The affinity of rNbSWP12 for phospholipids was assessed using a protein-lipid overlay assay; 0.5 µg/mL purified rNbSWP12 were incubated with PIP Strips™ membranes (Molecular Probes), immunoblotted with antibody against NbSWP12 and utilize ECL substrate (Bio-rad) for visualization. The rNbSWP12 protein can notably bind to Ptdlns(3)P.

**Figure 4 jof-08-00764-f004:**
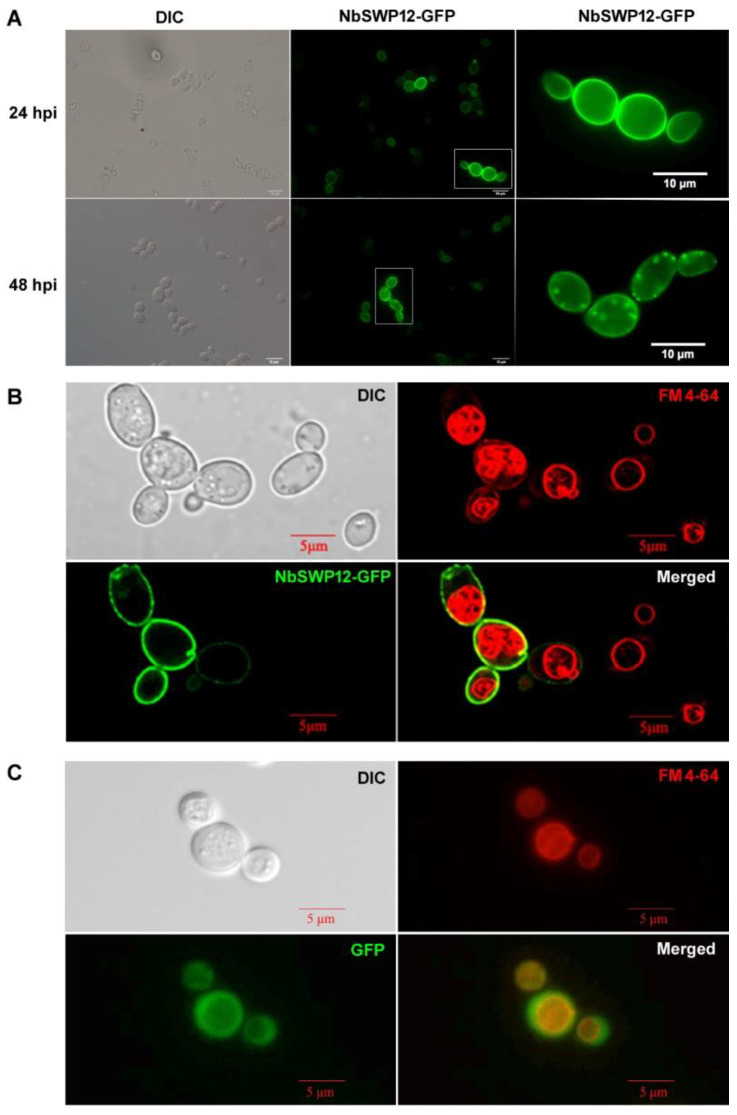
Localization of NbSWP12-GFP fusion protein expressed in yeast *S. cerevisiae* CEN.PK2. (**A**) Fluorescence microscopy images showed overexpressed NbSWP12-GFP targeted to cell membrane in the first 24 h of inducible expression and partial of NbSWP12-GFP located to endocytic vesicle after longer cultivation. (**B**) Different amounts of NbSWP12-GFP affected vesicle genesis. The yeast cells were stained with FM 4-64 for 1 h and observed after extra 3 h cultivation. Cells with low expression of NbSWP12-GFP showed large single vacuole, while numerous vesicles appeared in cells with plentiful expression of NbSWP12-GFP. (**C**) The *S. cerevisiae* CEN.PK2 (pUG35) was used as control. A large single vacuole was formed in the yeast that GFP was over-expression.

**Figure 5 jof-08-00764-f005:**
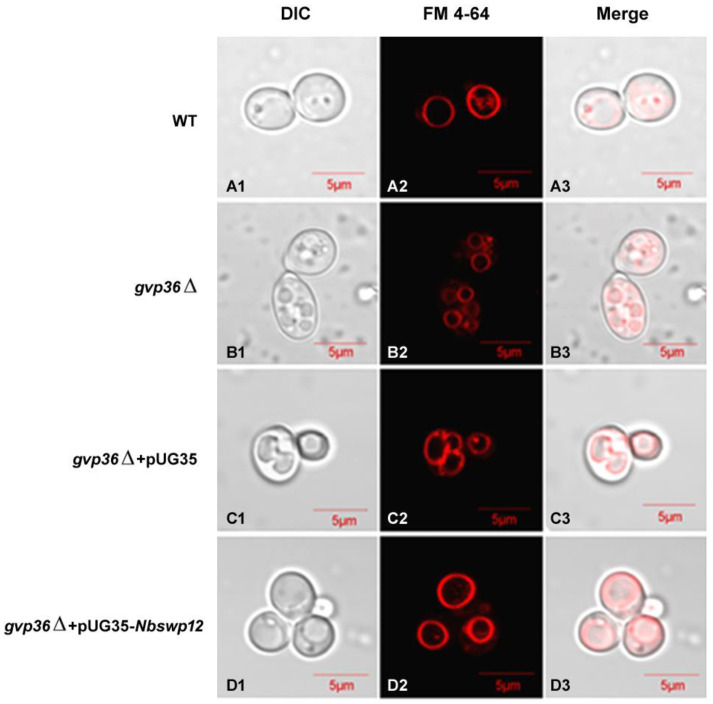
NbSWP12 rescued *gvp36*Δ mutant phenotype associated with vacuole biogenesis. WT—wild type (BY4742 strain); *gvp36*Δ—*gvp36* deletion mutant; *gvp36*Δ + pUG35—*gvp36*Δ strain containing pUG35 without an insert as control; *gvp36*Δ + pUG35- *Nbswp12*—*gvp36*Δ strain containing pUG35-*Nbswp12*; (**A1**–**D1**): Differential Interference Contrast; (**A2**–**D2**): FM 4-64—red fluorescent dye for vesicle staining; (**A3**–**D3**): Merged imagines.

**Table 1 jof-08-00764-t001:** Primers used in this study.

Plasmid	Primer	Sequence
pCold I-*Nbswp12*	*swp12-F*	CGGGATCCATGAAAGATTTTAAAAAGAA
pUG35-*Nbswp12*	*swp12-R*	GCGTCGACCTTAGTCCTCTCTAATGCTT
pGADT7-*Nbswp12*	*12AD-F*	GGAATTCCATATGATGAAAGATTTTAAAAAGAAAATT
	*12AD-R*	CGCGGATCCTTACTTAGTCCTCTCTAATGCTTT
pGBKT7-*Nbswp12*	*12BD-F*	CGCGGATCC ATGAAAGATTTTAAAAAGAAAATT
	*12BD-R*	AAAACTGCAGTTACTTAGTCCTCTCTAATGCTTT

## Data Availability

Not applicable.

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
