# Peer review of "Heterologous Expressed NbSWP12 from Microsporidia Nosema bombycis Can Bind with Phosphatidylinositol 3-Phosphate and Affect Vesicle Genesis"

_jof, 2022, doi:10.3390/jof8080764_

Round 1
Reviewer 1 Report
The paper by Chen et al. is important for the field of research on microsporidia, and on fungal biology in general. It contains novel information and opens a new perspective for future research. The authors demonstrate that one of microsporidian spore wall proteins belongs to BAR family of proteins responsible for membrane curvature and vesicle formation. As other proteins of this family, NbSWP12 produces dimers and binds to a membrane lipid, phosphatidylinositol 3-phosphate. On yeast heterologous system the authors managed to demonstrate that this protein not only was expressed in yeasts and targeted endosomal vacuoles, but its expression rescued the defects in vacuole biogenesis in S. cerevisiae BAR mutants. According to annotations of several published genome projects, higher microsporidia lack genes for endocytosis, and this adds an intrigue to the presented research. The methods used are adequate, and results are convincing. However, writing and presentation of the data needs additional work, mostly editorial, before the manuscript can be published. I enclose beneath my correction, suggestions and recommendations.
Abstract and Introduction
L16-17: The phrase is not well composed. Also “microsporidian” is an adjective; please, use here and elsewhere: “microsporidium” as a single noun; “microsporidia” as a plural, the Microsporidia as a phyllum; “microsporidian” as and adjective. Better to puy it like this:
I suggest the following version of this phrase: “Here, we demonstrated that the spore wall protein NbSWP12 of the microsporidium Nosema bombycis was a Bin/Amphiphysin/Rvs (BAR) protein that can specifically bind with phosphatidylinositol 3-phosphate, a lipid in the membrane with key functions in endosomal vesicle biogenesis and trafficking.”
L20: …. To the cell membrane and…
L22: I suggest: “Microsporidia (delete “are specific in nature, every species is) lack numerous gens in metabolism and vesicle trafficking. The study identified a BAR protein function in vesicle genesis and sorting, provided a clue (or clues) for further understanding how microsporidia internalize nutrients and metabolites during proliferation”.
L37. Comma after “rely on”; punctuation should be checked by the editor all over the mns
L 39: “microsporidia”, “are used” -not “was”
L41: lepidopterans, plural
L43-44: I suggest: “However, how endocytosis occurs in microsporidia and how the membrane of meronts migrates to form endocytic vesicles (delete) is unknown. May be: … and how the plasma membrane recycles during endocytosis(not “membrane migrates”)
L53: correct: ….where endosomes are generated
L55: commas before and after “like Snx9”
L61: “RNA interfere”???? Did you mean RNAi (interfering RNA?) or RNA interfered?
L62 delete “of”
L64-65: …..if it has a potential of participating in the endocytosis of materials (delete)
L66 microsporidia; and remove “And” at the beginning of the next sentence
L66-68: I suggest : “These parasites can only be cultured and proliferated in cell cultures, which also limits our ability to track their endocytic process. Therefore, yeasts Saccharomyces cerevisiae, a closely related model organism (yeasts are not “closely related to microsporidia”, so better delete this statement), were selected for further study of the protein….”
L69-70: Consider correcting: “In this study, we demonstrated that NbSWP12 had two features of BAR family, dimerization and lipids binding ability.”
L71: Consider correcting: “Lipid overlay assays demonstrated that NbSWP12 specifically bound to Ptdlns(3)P”.
L72: comma after “yeasts”
L73-74 Consider correcting: “This study opens a door for us (delete) to further understanding how microsporidia….”
Material and methods:
Detailed fluorescent/confocal microscopy section should be added to M&M, because visualization of GFP constructs in confocal microscope was in fact the major proof of the central hypothesis. What controls were used? What counterstain? As I see from fig 4, synaptic vesicle dye FM 4-64 stain applied; it is not mentioned in M&M.
Otherwise, I have no objections to the method section. Methods are adequate for the goal, well explained and up to date. However: the text must be carefully edited by the editor for English use, particularly for punctuation and grammar.
Results
L176 “microsporidia” here and elsewhere
The text of results and figure legends MUST be thoroughly edited by English-speaking editor for clarity and correct English use (articles, punctuation etc).
Discussion
No objection from the point of ideas and structuring. However, the authors demonstrate insufficient knowledge of the latest papers on genomics of microsporidia and related groups of Opishtosporidia. They cite only early paper by Katika et al., 2001. By the way, in opposite to the author's statement, genes for clathrin and adaptins were NOT found in higher Micosporidia to which N. bombycis belongs, but only in metchnikovellids, the basal lineage of Microsporidia (see Mikhailov et al, Galindo et al., Quandt et al) etc. It looks that endocytic pathway was largely eliminated in microsporidia in the course of evolution. This work suggests that some steps of this pathway have been retained… like maybe the ability to produce endocytotic vesicles from the plasma membrane. The discussion in this part should be a bit deeper and include analysis of the genomic data on microsporidia.
The narrative of Discussion needs attention and additional work. As it is, it can be considered only as a raw draft. It should be carefully edited taking in account my previous suggestions and corrections, preferably by or with the help of a native English speaker.
L.279 microsporidia
L. 280 sections, not slices
Reviewer 2 Report
Microsporidia is a unique group of unicellular eukaryotes closely related to (and monophyletic with) fungi but very different in biology due to intrinsic features acquired during evolutionary adaptation to intracellular parasitism. One of such outstanding features is extreme genome compaction accompanied with notable reduction of cellular machinery, including vesicle trafficking. This explains why the studies of molecular genetics of Microsporidia clarifying their interactions with the host cell are of particular interest to the readers of Journal of Fungi. The paper under review enlightens the crucial role of a specific protein in vesicle biogenesis and sorting using a series of approaches of molecular and cellular biology. The data presentation is clear and the conclusions are quite convincing. It can be agreed that the study opens new directions in microsporidia-host cell interaction mechanisms research.
The main drawback of the manuscript is poor grammar and style, to begin with the title where the first word should be the adverb, not the adjective.
The use of specific abbreviations is not welcome in the title as well and should be given in full, as well as at the first mentioning in the main body.
The information cited from previous works should not be repeated in introduction and discussion.
In Lines 306-307, the references are lacking.
“Microsporidia” is plural and should be used when the whole group or several species are implied; “microsporidium” is singular and should be used when a single species is mentioned; “microsporidian” is an adjective and should be used as such.
On the other hand, “lepidopteran” and “protozoan” are proper terms when a single species is mentioned but needs plural form for other cases.
The sentences in L16-17 (protein … was a … protein can bind), L34-35 (As a … microorganism …., the genome), L61 (RNA interfere), L123 (Primers were list), L223-224 (binding … was detected by immunoblotted… and utilize), L255-256 (cells … showed …, while …. expression exhibit), L261 (insure regular express), L277 (strategies such as directly interact), L278 (secrete molecules to regulate the host), L283 (in some of genome), L304 (at partial of vesicles) should be rephrased for clarity.
The sentences should not start with the conjunctions, such as “and”, as well as digits.
L24: “further understand how microsporidian internalize” = “further understanding how microsporidia internalize”
L34: “control of microsporidia” = “control of microsporidioses”
L70: “two feature” = “two features”
L74: “how microsporidium” = “how the microsporidium”/” how microsporidia”
L98: “cells which carries” = “cells which carry”
L105: “follow the instructions” = “following the manufacturer’s instructions”
L114: “for denature” = “for denaturation”
L120: “fragment … were” = “fragment was”
L159: “for overnight” = “overnight”
L159-160: “for overexress” = “for overexpression”
L161: “signal were” = “signal was”/”signals were”
L221: “can be interacted” = “can interact”
L246: “cell” = “cells”
L280 “slices” – do you mean “sections”?
Typos such as “gens” (L22), “obseved” (242), “genisis” (L258), and “paly” (L303) should be corrected
Author Response
Response to Reviewer 2’ Comments
- The main drawback of the manuscript is poor grammar and style, to begin with the title where the first word should be the adverb, not the adjective.
Response: We are very sorry for the linguistic problem in the previous version. Thanks to reviewers for revision suggestions, which improved the readability of the article. Besides, the language of the revised manuscript has been corrected and modified by a qualified person.
- The use of specific abbreviations is not welcome in the title as well and should be given in full, as well as at the first mention in the main body.
Response: Thanks for your suggestion. Ptdlns(3)P was changed to phosphatidylinositol 3-phosphate in the title. Full name with abbreviation was given at the first mention in the abstract and introduction.
- The information cited from previous works should not be repeated in the introduction and discussion.
Response: We modified and adjusted some references and reserved the necessary ones.
- In Lines 306-307, the references are lacking.
Response: The references were added in the revised manuscript.
- “Microsporidia” is plural and should be used when the whole group or several species are implied; “microsporidium” is singular and should be used when a single species is mentioned; “microsporidian” is an adjective and should be used as such.
Response: Thank you very much for your reminder, which is very important for us to use these terms correctly.
- On the other hand, “lepidopteran” and “protozoan” are proper terms when a single species is mentioned but needs plural form for other cases.
Response: “lepidopteran” and “protozoan” have been modified to “lepidoptera” and “protozoans”.
- The sentences in L16-17 (protein … was a … protein can bind), L34-35 (As a … microorganism …., the genome), L61 (RNA interfere), L123 (Primers were list), L223-224 (binding … was detected by immunoblotted… and utilize), L255-256 (cells … showed …, while …. expression exhibit), L261 (insure regular express), L277 (strategies such as directly interact), L278 (secrete molecules to regulate the host), L283 (in some of genome), L304 (at partial of vesicles) should be rephrased for clarity.
Response: The sentences have been rephrased in the revised manuscript.
- The sentences should not start with the conjunctions, such as “and”, as well as digits.
Response: It has been revised.
L24: “further understand how microsporidian internalize” = “further understanding how microsporidia internalize”
L34: “control of microsporidia” = “control of microsporidioses”
L70: “two feature” = “two features”
L74: “how microsporidium” = “how the microsporidium”/” how microsporidia”
L98: “cells which carries” = “cells which carry”
L105: “follow the instructions” = “following the manufacturer’s instructions”
L114: “for denature” = “for denaturation”
L120: “fragment … were” = “fragment was”
L159: “for overnight” = “overnight”
L159-160: “for overexress” = “for overexpression”
L161: “signal were” = “signal was”/”signals were”
L221: “can be interacted” = “can interact”
L246: “cell” = “cells”
L280 “slices” – do you mean “sections”?
Typos such as “gens” (L22), “obseved” (242), “genisis” (L258), and “paly” (L303) should be corrected
Response: We deeply appreciate your thorough review and careful modification. These corrections have been incorporated into the revised manuscript.
Reviewer 3 Report
This research is interesting, in which the gene functional assay in microsporidia is difficult. By using yeast system, the functional genomic of microsporidia can be resolved. However, I have one concern about the consensus of the mechanism between Saccharomyces cerevisiae and Nosema bombycis. Therefore, I suggested the RNAi experiment of NbSWP12 should be conducted in the N. bombycis infected silkworm to demonstrated the in-situ function of NbSWP12.
Minor points:
1. English writing should be checked.
2. L17: NbSWP12 of microsporidian Nosema bombycis “IS” a Bin/Amphiphy
3. L20, 93: Please make sure the protein name is italic or not (NbSWP12 and gvp36).
4. L56: Make sure the case of letters was true.
5. L109,139: The whole name of rNbSWP12 should be wrote when it first presented in the article.
6. L256: Has two dots.
7. L270: Wild type not wide.
Round 2
Reviewer 3 Report
The author did many experiments and addressed all my suggestions, therefore, I suggest this manuscript could be accepted by the journal.